# A Probabilistic Method for Estimating the Influence of Corrosion on the CuAlNi Shape Memory Alloy in Different Marine Environments

**Špiro Ivošević [1],\* , Nataša Kovač [2], Gyöngyi Vastag [3], Peter Majerič [4] and Rebeka Rudolf [4]**

[1] Faculty of Maritime Studies Kotor, University of Montenegro, I Bokeljske Brigade 44, 85330 Kotor, Montenegro

[2] Faculty of Applied Sciences, University of Donja Gorica, Oktoih 1, Donja Gorica, 81000 Podgorica, Montenegro; natasa.kovac@udg.edu.me

[3] Faculty of Sciences, University Novi Sad, Trg Dositeja Obradovića 3, 21000 Novi Sad, Serbia; djendji.vastag@dh.uns.ac.rs

[4] Faculty of Mechanical Engineering, University of Maribor, Smetanova ul.17, 2000 Maribor, Slovenia; peter.majeric@um.si (P.M.); rebeka.rudolf@um.si (R.R.)

\* Correspondence: spiroi@ucg.ac.me; Tel.: +382-67-628-985

**Abstract:** This paper gives an approach to the probabilistic percent corrosion depth estimation model for the CuAlNi Shape Memory Alloy (SMA) in different marine environments. Real testing was performed for validation of the theoretical model, where CuAlNi SMAs were exposed to 6 and 12 months in different seawater environments. Focus Ion Beam (FIB) analysis was used to measure the real corrosion depth on the surfaces of tested samples. A statistical approach to the investigation of the corrosion rate of CuAlNi SMA is given, where the corrosion rate is observed as a continuous random variable described by a linear corrosion model, with the assumption that corrosion starts immediately upon alloy surfaces being exposed to the influences of the marine environment. The three best-fitted two-parameter distributions for estimating the cumulative density function and the probability density function of the random variable were obtained by applying adequate statistical tests. Furthermore, using EDX analyses, we identified the chemical composition of the corroded materials, and with the help of Principal Component Analyses, we determined which corrosion environment had the most dominant influence on the corrosion process. The research results indicated that the changeable environment in the tides had a more heterogenic chemical content, which accelerated the corrosion rate.

**Keywords:** CuAlNi Shape Memory Alloy; corrosion rate; probabilistic method; Focus Ion Beam analysis; two-parameter distributions

## 1. Introduction

Although in the last century, special attention was focused on smart materials based on the research of alloys, polymers, ceramics, composites, and hybrid systems, the most significant discoveries are related to Shape Memory Alloys (SMA). Since 1932, when the Swedish physicist who determined that gold–cadmium (Au–Cd) alloys could be deformed plastically when cool, and returned to their original shape when heated, a lot of research has been conducted [1]. Later in 1938, Greninger and Mooradian [2] first observed the Shape Memory Effect (SME) for copper–zinc (Cu–Zn) alloys and copper–tin (Cu–Sn) alloys, and in 1959 [3], the NiTi alloy was discovered by William Buehler, and the potential commercialization of NiTi was available from 1962 [3], which was known as nitinol [4].

Further, numerous research was performed to investigate its key thermo-mechanical SMA properties such as SME, superelastic and pseudoelastic effects, high damping capacity, and double SME.

Considering the basic mechanical, chemical, and characteristics of alloys, as well as their characteristic shape memory, they have a wide variety of applications for engineering and technical applications, such in medicine, aerospace, vibration control, biomechanics, automotive, marine, robotic and different domestic applications. SMAs were applicable for blood clot filters, orthodontic corrections in biomedicine [5], eyeglass frames, antennas of mobile phones, bra underwires for domestic application, fluid connectors, coupling and thermal actuators, fire safety valves, and electric circuit breakers in industrial applications [6], pipe and tube couplings in the aircraft industry, sensors and thermal or electrical actuators in automobiles and connectors in robotic arms [6].

Numerous alloys have been investigated to date in order to improve Shape Memory Alloys' thermomechanical properties. Furthermore, we can consider about 20 elements in the central part of the Periodic Table whose alloys exhibit SME. Binary, trinary, and fourth systems of alloys were investigated, but mainly three alloy systems were the focus of research and development of the shape memory phenomena based on NiTi, Cu, and Fe.

NiTi Shape Memory Alloy is one of the more functional, successful, commercial, and useful alloys. Furthermore, the NiTi SMAs are biocompatible, exhibit high wear resistance, are thermally stable, and show excellent shape memory strain up to 8% [7]. NiTi is generally deployable up to 80 °C, thus extensive research has been devoted to it by developing new alloy compositions in order to increase their transformation temperatures above 100 °C [7]. In that sense, ternary NiTi alloys were investigated by adding Pd, Pt, Hf, or Zn. However, their commercial applications are limited due to some of their disadvantages, such as low transformation temperatures, difficulty in production and processing, complexity, and cost.

Among the different Cu-based SMAs, CuAlNi alloys have a higher thermal stability (around 200 °C) than CuZnAl, CuAlBe, and CuAlMn alloys (maximum 120 °C) [5], while the CuZnAl alloys show better ductility as compared to CuAlNi alloys for low temperature applications [7]. Furthermore, CuAlNi alloys have low production cost, better machinability, better work/cost ratio, are easier to manufacture, and have a higher range of potential transformation temperatures [8]. The SME of CuAlNi alloys is able to display, at a specific composition, about 11–14 wt.% for aluminum and 3–5 wt.% for nickel [5].

However, the base alloys suffer from bad cold workability and martensite stabilization; hence, ternary quaternary additions in various amounts have been tried by different investigators to improve upon the properties and remove the drawbacks [9]. The biggest disadvantage of polycrystalline CuAlNi alloys is their small reversible deformation (one-way memorized effect, up to 4%, and two-way memorized effect: Only around 1.5%), which occurs thanks to intergranular fracture already at low average stress levels [10]. However, CuAlNi SMAs suffer from high brittleness, which is associated with the large elastic anisotropy and large grain size. As a result, many researchers have tried to refine the grain size of Cu-based SMAs through the addition of alloying elements and/or applying different thermal aging treatment conditions [10].

In many articles, additional alloying elements were analyzed with CuAlNi alloys, such as Zr and Ti or Mn and B, in order to investigate the microstructure and mechanical properties of this alloy system [5]. It has been shown that the poor workability of CuAlNi alloy caused by brittle intergranular fracture can be reduced by refining the microstructure through the addition of Co, Mn, Ti, or Zr [7]. Rapid solidification constitutes an alternative route for grain refinement and the suppression of brittle phases in Cu-based SMAs [7].

Metal materials are affected mainly by electrochemical corrosion when in the marine environment, or the atmosphere, that is affected by the sea. In these conditions, the corrosion rate and the change in chemical composition caused by corrosion depend mainly on the conductivity of seawater, wind, sea waves, salinity, ph, temperature, and other environmental factors. Corrosion rate and the change in the chemical composition will vary if a metal material is exposed to the marine environment, atmosphere, and changeable conditions at the sea surface directly. This is caused by the merging of the sea and atmosphere due to waves and ebb and flood tides.

## 1.1. Shape Memory Alloys in Marine Enviroments

Although the application of SMAs in marine and maritime applications is not as common as in medicine and the transport industry, different potential applications can be found from the deep sea to the sea surface. New SMAs can be used for connecting tubes in the deep sea due to their superelasticity properties. SMA-based tendons may potentially resolve problems in tension-leg platform structures, SMA thermostats for subsea equipment, and transferring thermal energy in electricity under the power plant [11].

The application of SMAs in marine and maritime applications will certainly depend not only on the mechanical, chemical, physical, and thermo-mechanical characteristics of the material but also on their resistance to the specific environmental conditions in which they are located, and, above all, to corrosion. From the literature, it was observed that the current density for samples in austenitic structures is much greater than for martensitic structures in SMAs. This demonstrates that SMAs have more corrosion resistance than traditional alloys due to the hyperplastic behavior of their polycrystalline structure [12].

For porous NiTi SMAs, the corrosion rate is high due to their large surface area and specific surface morphology, while stress corrosion causes cracking in NiTi alloys [13].

In the research on the CuZnNi alloy, where the wt% of the Ni content was increasing from 2–9, or increasing the zinc concentration in alloy composition, it should be concluded that increases in the corrosion resistance property of the alloy also increases in the three different corrosive mediums—freshwater, hank's solution, and seawater [13].

The addition of aluminum to the Cu-based alloys increases its corrosion resistance due to the formation of a protective layer of alumina, while the presence of nickel is important in the passivation of Cu–Ni alloys because of its incorporation in the Cu(I) oxide, which is formed on the corroded surface of the alloy [5].

In the article where the influence of different chloride ion concentrations (0.1%, 0.5%, 0.9%, and 1.5% NaCl solution) on the electrochemical behavior of the cast CuAlNi alloy was examined, polarization measurements revealed that an increase in chloride ion concentration leads to an increase of the corrosion current density values and a decrease of the polarization resistance values, which indicated a higher corrosion attack on the alloy [8].

## 1.2. About the Corrosion Rate of Metal Structures in Marine Enviroments

Corrosion accelerates the decay of a metal that is exposed to various influences of the environment significantly. Previous research focused on the development of the models of pitting and general corrosion, which was described on the basis of the depth and rate of corrosion, as well as the conditions which cause corrosion. The application of new materials in the past typically required laboratory testing or a shorter examination in nature due to the inadequate observation of the materials over the past decades. These models predicted the probability of the emergence of corrosion through the identification of key variables and corrosion mechanisms. There are also statistical models, which gather data on corrosion of the structures in exploitation and then calculate the average and standard deviation of corrosion rate for metal structures. Numerous studies applied this model in the examination of the decay of structural materials on ships [14–19].

Figure 1 presents the available corrosion models in accordance with the existing research results and the analysis of corrosion development over time. Corrosion does not emerge on the metals protected with surface coatings, even in a corrosive environment. Corrosion occurs after the wear and cracking of coatings, which can be illustrated by the different (b and c) curves in Figure 1. The majority of authors consider corrosion an unstable, time-dependent process, whose rate can be expressed linearly (Figure 1, curve a.) However, experimental research confirmed that nonlinear models describe corrosion processes better in certain environmental conditions (Figure 1., b and c curves).

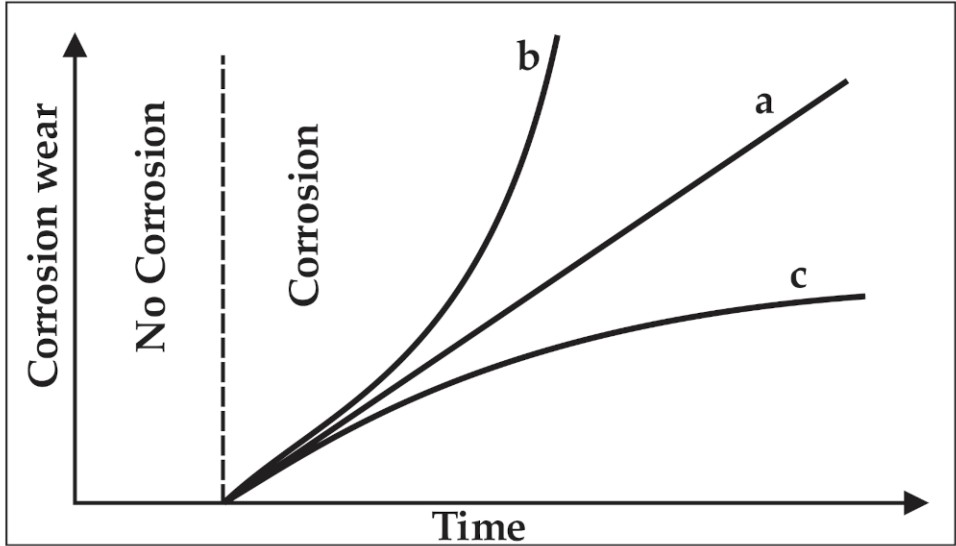

**Figure 1.** General model of corrosion rate.

The graphic view of a corrosion process that is increased and accelerated over time will be concave (b curve in Figure 1) and correspond to the conditions of immersed/submerged structures. This is especially relevant for the structures with dynamic strain whose metals are exposed to corrosion constantly. The corrosion, which is initially accelerated and subsequently slower, is best illustrated by the c curve in Figure 1. This process is characteristic for the structures which are not submerged in the sea and whose metals are notably covered by a corrosive layer that prevents further exposure of the metal to the corrosive environment.

The research on the influences of corrosion on different vessels proves that exposure to the atmosphere, operating factors of vessels, as well as various biological, chemical, and physical factors of seawater, can accelerate the corrosion of metal materials. Furthermore, if we analyze the main marine environment factors such as dissolved oxygen, pH, temperature, water movement, salinity, sulfate-reducing bacteria, galvanic coupling, marine growths, and their influence on the rate and mechanism of corrosion, we can conclude that they are strongly correlated. Changing any of the mentioned influences usually produces changes in others. As a result, the relationships between site, artifact condition, and corrosion product are often complex, and it is important to analyze it in order to find any specific correlation. All in all, it is important to predict the corrosion of metal structures before they are used for any type of marine application.

## 2. Materials and Methods

Therefore, this research focused on the investigation of CuAlNI SMAs characteristics and their corrosion resistance in maritime environments. The analysis encompassed 3 types of samples that were exposed to the sea, atmosphere, and tidal zones for 6 months and 3 samples that were exposed to the same conditions for 12 months. In order to determine corrosion rate and changes in chemical composition on the surface of CuAlNi SMA samples, 2 databases were used for the verification of research results. The 1st database was related to corrosion depth on the surface of the CuAlNi samples, which was expressed in nm and determined by means of the Focused Ion Beam (FIB) method. The 2nd database was related to the changes in chemical composition, which were determined by an Energy Dispersive Spectrometer (EDX) analysis.

### 2.1. Materials
2.1.1. Production of CuAlNi

CuAlNi SMA bars were produced with the continuous casting process with a laboratory scale vertical continuous casting device, Technica Guss, which was connected to

a 60 kW medium-frequency (4 kHz) Vacuum Induction Melting (VIM) furnace, Leybold Hereaus. The withdrawal parameters were programmable, thus an almost arbitrary time-velocity curve can be realized (the limits being set by the performance of the motor and inertia of moving parts). In that sense, different methods were used for this research [20].

Pure metals were used for the production of CuAlNi SMA bars: Cu (99.99 wt.%), Al (99.99 wt.%) and Ni (99.99 wt.%) delivered by Zlatarna Celje d.o.o. Slovenia. The bars were then cut by electro-erosion into test samples with selected dimensions of 2r = 7 mm (6 pieces) (see Figure 2), which corresponded to the corrosion tests. All test samples were ground after electro-erosion to remove erosion residues and impurities and then were prepared metallographically for initial microstructure observation for corrosion testing.

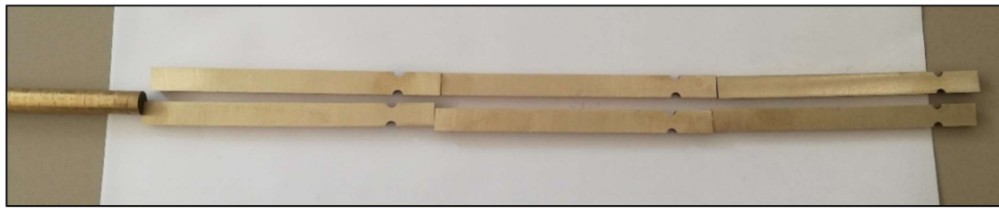

**Figure 2.** Presentation of CuAlNi casting and cut-off samples for corrosion testing.

### 2.1.2. Preparation of CuAlNi Samples for Microstructure Observation

To monitor the occurrence of the corrosion process, it was necessary to perform a microstructure analysis of the initial state cross-section. Based on this, 1 sample was mounted in a hot-mounting mass and ground with abrasive paper in grades of 180–4000 on the grinding/polishing machines BUEHLER Automet 250 and EcoMet 250. The sample was polished with a napless cloth and polishing suspension with $Al_2O_3$ with the size of 1 μm. After polishing, the sample was cleaned by ultrasound. This process was followed by etching in order to reveal the microstructure of the sample [21].

Before performing corrosion tests, samples were taken using an Optical Microscope microstructure overview, microhardness measurement of the test samples (Figure 3a–c). The chemical composition of CuAlNi was measured using Inductively Coupled Plasma Optical Emission Spectroscopy—CP–OES (Agilent 720, IMT, Ljubljana, Slovenia) and X-Ray Fluorescence (XRF) with a Thermo Scientific Niton XL3t GOLDD + XRF Analyzer. The compositions of the CuAlNi samples were obtained, as shown in Table 1 [21].

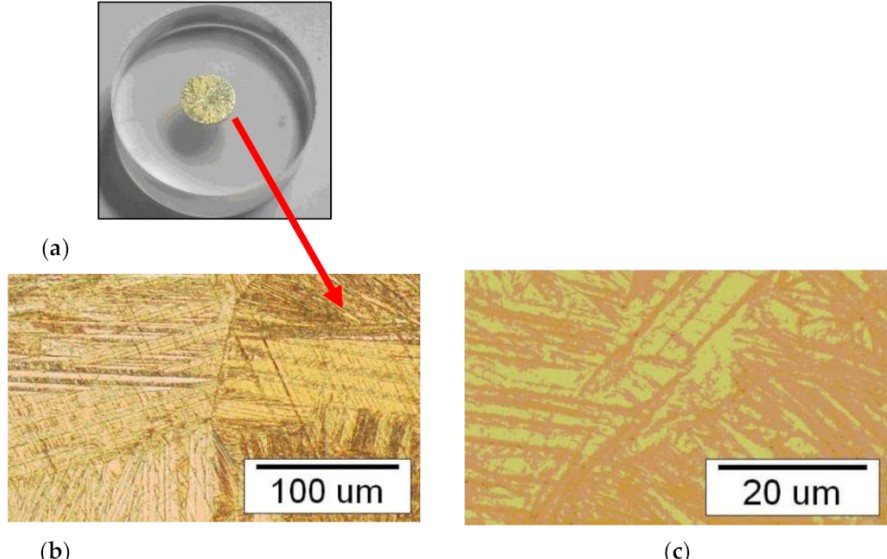

**Figure 3.** CuAlNI alloy: (**a**) Cross-section of the testing sample, (**b**,**c**) typical microstructure.

**Table 1.** Percentage composition of the studied CuAlNi alloy.

| Sample | % Cu | | % Al | | %Ni | | %Fe | |
|---|---|---|---|---|---|---|---|---|
| | ICP-OES | XRF | ICP-OES | XRF | ICP-OES | XRF | ICP-OES | XRF |
| CuAlNi | base | base | 12 | 9.4–9.6 | 3.9 | 4.4 | 0.03 | |

*2.2. Proposed Problem and Related Methodology*

The experiment was based on the observation of the CuAlNi alloy in 3 different locations. In the 1st location, 2 samples were positioned near the sea, 3 m above the sea surface, and thus exposed to the influences of the atmosphere and marine environment. The 2nd set of samples was positioned at the sea surface and, thus, exposed to the changeable influences of the atmosphere and the sea, depending on the ebb and flood tides, as well as waves (flushing). The 3rd set of samples was immersed into the sea at a depth of 3 m near the coast.

The paper relied on 2 directions of data analysis, as it is graphically presented on Figure 4. In the 1st direction, on the basis of Focused Ion Beam characterization were measuring corrosion depth in nm on the surface of CuAlNi after 6 and 12 months and applying an appropriate linear corrosion model in order to find the best 2 parameter distribution functions. In the 2nd direction, using Semi-quantitative Analysis, we obtained the chemical composition of the alloy surface and applied Principal Component Analysis in order to discover the influence of the different environments on the change in the chemical composition of the material.

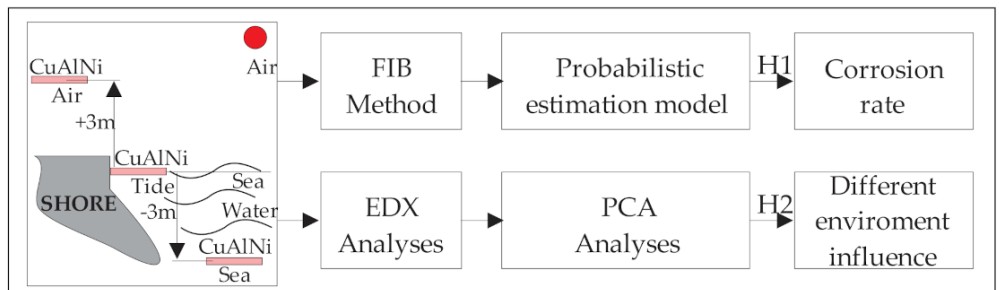

**Figure 4.** The scheme of the conceptual model of the research for CuAlNi alloy.

2.2.1. Focused Ion Beam (FIB) Description

Material characterization by the FIB/SEM method (Focused Ion Beam on a Scanning Electron Microscope) allowed simultaneous physical processing of materials with a Focused Ion Beam and imaging of the samples below the surface. The FIB/SEM used for the cross-section milling and upper layer measurements of the samples were a Quanta 200 3D (FEI, USA) with a gallium ion source.

2.2.2. Semi-Quantitative EDX Analysis

The chemical composition of the selected CuAlNi was determined through the use of a high-resolution Field Emission SEM Sirion 400 NC (FEI, Watertown, MA, USA), equipped with an EDX detector—INCA 350 (Oxford instruments, Abingdon, Oxfordshire, UK). The EDX semi-quantitative analysis determined the chemical composition of the materials after corrosion, as well as the content of elements on the surface of the examined samples. In that way, the chemical composition of the CuAlNi surfaces was identified for each selected sample, up to several spectrums per sample under different magnifications.

2.2.3. Probabilistic Corrosion Rate Estimation Model

In order to undertake investigations of the analytic and probabilistic corrosion rate estimation model for different CuAlNi samples, which were located in 3 different locations,

we referred to the survey paper by Qin and Cui [22]. It is well known that the wear of plate thickness, $d(t)$, due to corrosion may generally be expressed as a power function of the time (usually expressed in years or months and in mm or nm) after the corrosion starts (see, e.g., [22,23]), i.e.,

$$d(t) = c_1(t - T_{cl})^{c_2}, \tag{1}$$

where $d(t)$ is the corrosion wastage expressed in nanometers (nm); $t$ is the elapsed time after the plate is used; $T_{cl}$ is the life of the coating; $c_1$ and $c_2$ are positive real coefficients. This model was proposed in [23]. The coefficient $c_2$ may usually be assumed to be $1/3$, or, pessimistically, assumed to be 1, while the coefficient $c_1$ is indicative of the monthly corrosion rate. As noticed in [22], in most of the studies on time-dependent reliability of ship structures (see, e.g., [24,25]), the effect of corrosion was represented by an uncertain but constant corrosion rate, which resulted in a linear decrease of plate thickness with time, in spite of several authors establishing that some nonlinear models were more appropriate.

The validity of the expression (1) with $c_2 = 1$ proposed by Paik, Kim, and Lee [14] was verified in this work. It is assumed that $T_{cl} = 0$ months, i.e., $d(t) = c_1 t$. This value was chosen due to the fact that the samples were not treated with anti-corrosion coating, which means that corrosive processes began immediately after exposure to the seawater environment, resulting in $T_{cl} = 0$ [17]. In order to determine the approximate value of $c_1$, the values of averages of the depth of corrosion $d(t)$ at time $t$ were used for the CuAlNi alloy exposed to the different seawater environments.

### 2.2.4. Principal Component Analyses

Multivariate methods, such as Cluster Analysis (CA) and Principal Component Analysis (PCA) were some of the most used mathematical methods for qualification and classification of voluminous and heterogeneous experimental data and determining the relationships between them [26–29].

These methods were considered suitable, as they enabled successful classification of a large number of data of different origins, as well as the identification and elimination of redundant information. Chemometric calculations were made by the Statistics 13.5.017 software (StatSoft Inc., Tulsa, OK, USA). The Origin 6.1 software was used for the processing of the obtained experimental results.

For this research, PCA was performed on a matrix, in which the experimentally obtained corrosion parameters from the EDX analyses were variables (columns), while the different parts on the observed alloy samples (spectrums) represent rows. The matrix data were standardized before the calculation in order to ensure the equal importance of all the analyzed parameters.

### 2.3. Data Collecting Analysis

The experiment with the CuAlNi samples was conducted in 3 different locations and during the years 2018 and 2019. An adequate evaluation of the influences of the seawater environment on corrosion was required to conclude relevant environment parameters for the Bay of Kotor and observed over a long period of time prior to the research.

Average temperatures of the sea and atmosphere were below 20 °C during the period of observation, while the average monthly temperatures of the sea were higher than the average temperatures of the air. The difference between temperature values varied between 0.8 °C for August and 10.3 °C for December [30]. Likewise, the maximum temperatures of the sea were considerably lower than the maximum monthly air temperatures, which varied between 0.2 °C in December and 10.9 °C in March, respectively. The minimum sea temperatures, on the other hand, were significantly higher than the minimum air temperatures, which varied between 12.0 °C in October and 22.6 °C in December. This indicated notably lower aberrations in the sea temperature in comparison with the air temperature [31].

The data about seawater temperature, conductivity, and salinity show that there were no significant aberrations in the values obtained on the sea depth up to 5 m. The average

temperature was 18 °C, conductivity was between 44.29 and 47.45, and salinity between 30.83 and 28.79, respectively [31].

The salinity and conductivity of the sea decreased on the surface between September and May due to the rainy season and the inflow of freshwater. Compared to the atmosphere collected temperature data, higher sea temperatures in the period observed and other influences of the sea (salinity and conductivity) rendered the corrosion processes in the sea significantly faster than the corrosion processes in the atmosphere.

Every CuAlNi surface of all samples was scanned under a particular magnification. Figure 5 shows a sample after 12 months of exposure to the sea, along with the corrosion depth on the alloy surface expressed in nm. Other samples exposed to the air, sea, and tidal zone were scanned in a similar way. Table 2 shows the data on magnifications, as well as corresponding corrosion depth and average values after 6 and 12 months of exposure.

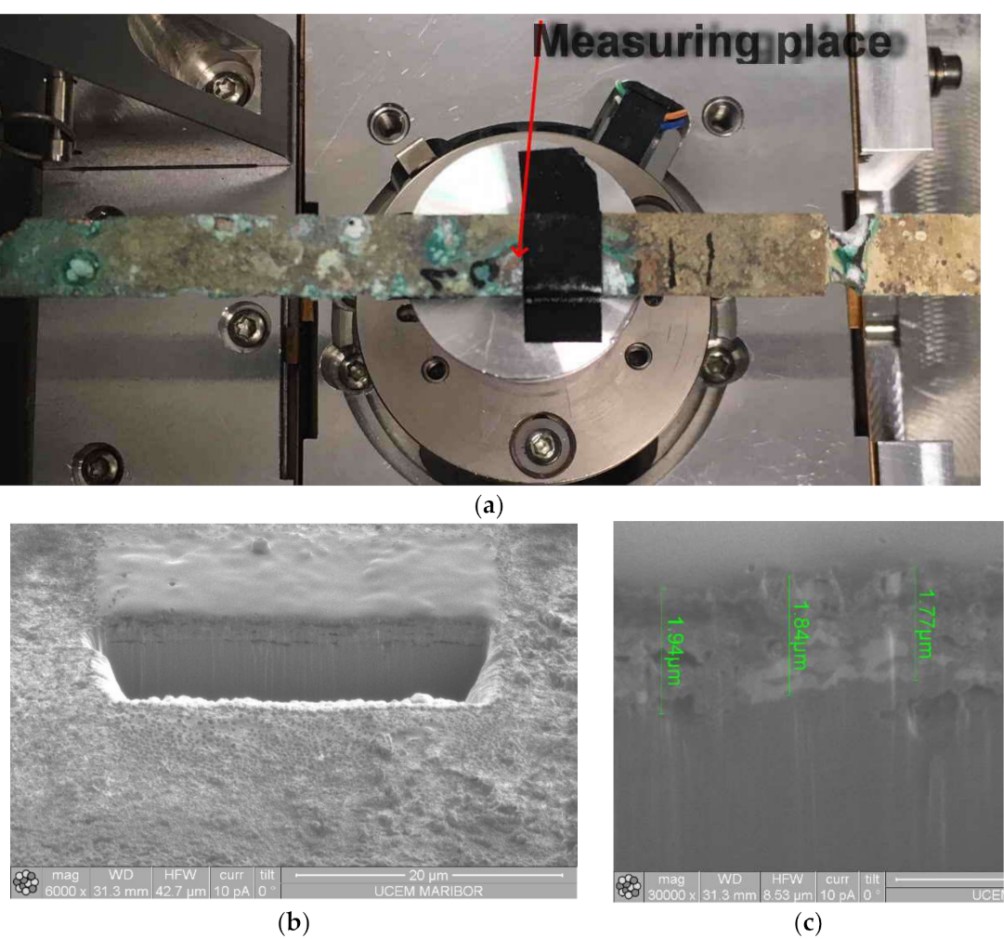

**Figure 5.** Focus ion beam (FIB) analyses, (**a**) place 1 of analysis, (**b**) photo view of sample, (**c**) measurement value of the corrosion layer in nm on a sample 1 year in the sea (sample C with 30,000 magnification, Table 2b).

**Table 2.** Corresponding corrosion wear for air/tide/sea CuAlNi alloy: (**a**) After 6 months, (**b**) after 12 months.

(**a**)

| | Corrosion Layer (nm) | | | | | Average Value (nm) |
|---|---|---|---|---|---|---|
| **AIR** | **Item-1** | **Item-2** | **Item-3** | **Item-4** | **Item-5** | |
| A [1] (×30,000) | 300 | 158.33 | 175 | 183.33 | 241.67 | 211.66 |
| B [1] (×30,000) | 225.15 | 250 | 183.33 | 225 | 275 | 231.69 |
| C [1] (×60,000) | 341.67 | 273.5 | 266.67 | 295.83 | 308.36 | 297.20 |
| D [1] (×60,000) | 216.67 | 162.5 | 175 | 262.5 | 220.83 | 207.5 |
| **TIDE** | | | | | | |
| A [1] (×30,000) | 1260 | 1260 | 1420 | 958.33 | 1090 | 1197.66 |
| B [1] (×30,000) | 575 | 833.33 | 825 | 825 | 1220 | 855.66 |
| C [1] (×30,000) | 1690 | 1720 | 1970 | 1670 | 2120 | 1834 |
| D [1] (×60,000) | 545.83 | 545.83 | 583.33 | 866.67 | 775 | 663.33 |
| **SEA** | | | | | | |
| A [1] (×20,000) | 787.5 | 812.5 | 812.5 | 900 | 662.5 | 795 |
| B [1] (×30,000) | 850 | 991.67 | 558.33 | 741.67 | 733.33 | 775 |
| C [1] (×30,000) | 758.93 | 946.43 | 785.71 | 482.14 | 553.57 | 705.35 |
| D [1] (×60,000) | 816.67 | 879.17 | 883.33 | 891.67 | 787.5 | 851.66 |

[1] A, B, C, and D are samples' photo views with different magnification of 20,000, 30,000, 60,000.

(**b**)

| | Corrosion Layer (nm) | | | | | | | Average Value (nm) |
|---|---|---|---|---|---|---|---|---|
| | **Item-1** | **Item-2** | **Item-3** | **Item-4** | **Item-5** | **Item-6** | **Item-7** | |
| **AIR** | | | | | | | | |
| A [1] (×30,000) | 191.67 | 258.33 | 691.67 | 166.67 | 641.67 | 133.33 | 475 | 365.48 |
| B [1] (×30,000) | 1200 | 1110 | 725 | 575 | 791.67 | 175 | 216.67 | 684.76 |
| C [1] (×30,000) | 150 | 125 | 675 | 575 | 783.33 | 891.67 | 975 | 596.43 |
| D [1] (×60,000) | 120.83 | 187.5 | 187.5 | 116.67 | 145.83 | 116.67 | | 145.83 |
| **TIDE** | | | | | | | | |
| A [1] (×30,000) | 2510 | 1620 | 2270 | 2530 | 2540 | 3330 | | 2466.67 |
| B [1] (×30,000) | 2670 | 2020 | 1890 | 1760 | 3050 | | | 2278.00 |
| C [1] (×30,000) | 1250 | 2170 | 2020 | 1810 | 2350 | | | 1920.00 |
| D [1] (×60,000) | 1210 | 2190 | 2390 | 2130 | 1810 | | | 1946.00 |
| **SEA** | | | | | | | | |
| A [1] (×20,000) | 1970 | 2150 | 1970 | 1950 | 1800 | | | 1968.00 |
| B [1] (×30,000) | 2120 | 2040 | 2160 | 2200 | 2240 | | | 2152.00 |
| C [1] (×30,000) | 1940 | 1840 | 1770 | 1940 | 1890 | | | 1876.00 |
| D [1] (×60,000) | 2030 | 2010 | 2040 | 2010 | 2110 | | | 2040.00 |

[1] A, B, C, and D are photo views with different magnification of 20,000, 30,000, 60,000.

## 3. Results

### 3.1. EDX Results

The second investigation included the systematization of the data from the EDX analysis in order to obtain the presence of the chemical elements on the surface of the CuAlNi samples, separately for all samples.

Figure 6 and Table 3 show the chemical composition of the CuAlNi samples that were exposed to the influences of the sea for 12 months. The data presented (from 1 to 3) as the weight percentage of elements, Mean, Standard Deviation, minimum and maximum values. The lower detection limits of EDX analysis were considered to be 0.1 wt%, with a relative

uncertainty of ±2% for major constituents with a mass fraction greater than 10 wt%, when using contemporary EDX spectrum processing techniques [32]. The error distribution increased rapidly for minor constituents (<10 wt%), where the relative uncertainty may be considered to be up to ±25% for commercial fitted standards procedures and up to ±50% for standardless procedures [33]. The analysis errors were considered to be relevant for the given investigation of the surfaces of samples after their exposure to seawater and the surrounding environments.

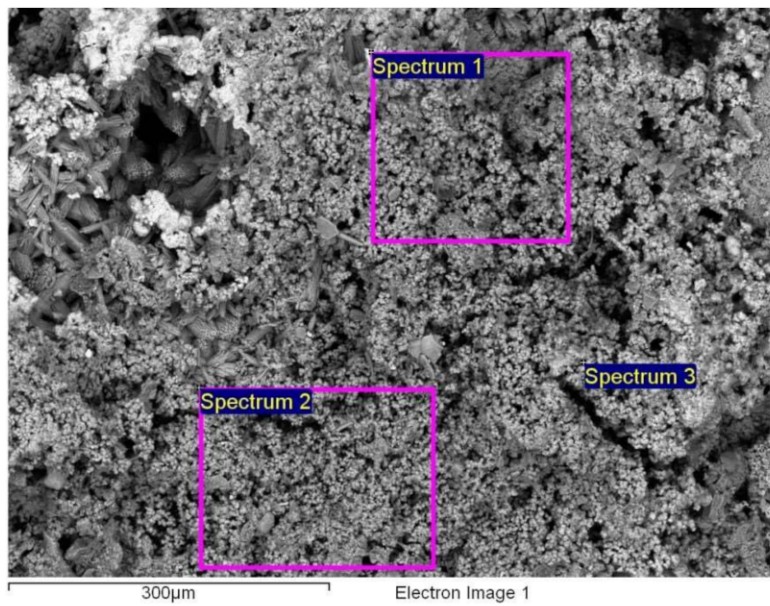

**Figure 6.** CuAlNi sample exposed after 12 months in the sea.

**Table 3.** Chemical composition in wt%.

| Spectrum | C | O | Mg | Al | Si | S | Cl | Ca | Cu | Total |
|---|---|---|---|---|---|---|---|---|---|---|
| Spectrum 1 | 14.58 | 53.11 | 5.56 | 5.79 | 2.27 | 1.39 | 0.36 | 3.03 | 13.90 | 100.00 |
| Spectrum 2 | 12.51 | 52.38 | 5.93 | 7.03 | 3.36 | 0.87 | 0.55 | 1.97 | 15.40 | 100.00 |
| Spectrum 3 | 12.01 | 44.27 | 7.46 | 6.97 | 0.26 | 0.51 | 2.64 | 0.38 | 25.50 | 100.00 |
| Mean | 13.04 | 49.92 | 6.32 | 6.59 | 1.96 | 0.92 | 1.18 | 1.80 | 18.26 | 100.00 |
| Std. Dev. | 1.36 | 4.91 | 1.01 | 0.70 | 1.57 | 0.44 | 1.26 | 1.33 | 6.31 | |
| Max. | 14.58 | 53.11 | 7.46 | 7.03 | 3.36 | 1.39 | 2.64 | 3.03 | 25.50 | |
| Min. | 12.01 | 44.27 | 5.56 | 5.79 | 0.26 | 0.51 | 0.36 | 0.38 | 13.90 | |

In the same way, each of the six samples underwent EDX analysis, and a specific number of spectrums was observed for each sample. Table 4 (a and b) shows the specific numbers of samples and spectrums for all the CuAlNi alloys samples exposed to different seawater environments after 6 months and 12 months.

**Table 4.** The number of samples and spectrums for: (**a**) The CuAlNi alloys in the sea, tidal zone, and atmosphere after 6 months' exposure, (**b**) the CuAlNi alloys in the sea, tidal zone, and atmosphere after 12 months' exposure.

| | | | | CuAlNi | | | | | |
|---|---|---|---|---|---|---|---|---|---|
| | **Air** | **Magn.** | **No. of Spec.** | **Tide** | **Magn.** | **No. of Spec.** | **Sea** | **Magn.** | **No. of Spec.** |
| (a) 6 months' exposure | Sample 1 | 200 | Spec 1–6 | Sample 1 | 200 | Spec 1–6 | Sample 1 | 200 | Spec 1–6 |
| | Sample 2 | 100 | Spec 1–6 | Sample 2 | 100 | Spec 1-6 | Sample 2 | 200 | Spec 1–6 |
| | Sample 3 | 70 | Spec 1–6 | Sample 3 | 70 | Spec 1–7 | Sample 3 | 100 | Spec 1–6 |
| | | | | | | | Sample 4 | 100 | Spec 1–4 |
| | | | | | | | Sample 5 | 70 | Spec 1–7 |
| (b) 12 months' exposure | Sample 1 | 300 | Spec 1–8 | Sample 1 | 300 | Spec 1–6 | Sample 1 | 300 | Spec 1–3 |
| | Sample 2 | 200 | Spec 1–7 | Sample 2 | 200 | Spec 1–6 | Sample 2 | 100 | Spec 1–3 |
| | Sample 3 | 100 | Spec 1–6 | Sample 3 | 100 | Spec 1–6 | Sample 3 | 200 | Spec 1–3 |

### 3.2. The Appropriate Statistical Analysis of Corrosion Rate Related to the CuALNi Alloy

Considering the conceptual model shown in Figure 4, the following two subsections will be shown the results in two specific directions of research. The first is corrosion depth and the second is changes in the chemical composition of the alloy due to different environment and exposure time influences.

Statistical analysis was conducted on the basis of 47 measurements of the depth of corrosion caused by the influences of the air (20 measurements were performed after 6 months and 27 measurements after 12 months); 41 samples whose depth of corrosion was caused by flood tide (20 measurements were performed after 6 months and 21 measurements after 12 months) and 40 measurements of the depth of corrosion caused by the influence of seawater (20 measurements were performed after 6 months and 20 measurements after 12 months). Table 5 shows the basic characteristics of the measured data.

**Table 5.** Descriptive statistics for corrosion caused by the influence of different environments.

| | **6 Months** | | **12 Months** | |
|---|---|---|---|---|
| | **Mean** | **Standard Deviation** | **Mean** | **Standard Deviation** |
| Air | 237.02 | 53.97 | 459.32 | 346.31 |
| Tide | 1137.67 | 489.56 | 2167.62 | 522.78 |
| Sea | 781.76 | 132.85 | 2009.00 | 129.16 |

Since the samples were not treated with anti-corrosion coatings, the initial hypothesis was that corrosion started immediately after the exposure of samples to the corrosive influences of the environment. It was assumed, therefore, that $T_{cl}$ equals 0 ($T_{cl} = 0$). The approximate value of the monthly corrosion rate—C1 coefficient—was determined on the basis of the mean values from Table 5. The linear model for $d(t)$ was determined by means of a linear regression, which was performed by the Wolfram Mathematica 9 software. Assuming that corrosion emerges immediately after the exposure of samples to environmental factors, the linear model for the depth of corrosion caused by the influences of the air, tide, and the sea was labeled as $d_a(t), d_t(t), d_s(t)$ and expressed as:

$$d_a(t) = 38.522t, \tag{2}$$

$$d_t(t) = 182.43014t, \tag{3}$$

$$d_s(t) = 148.99187t, \tag{4}$$

Figure 7 shows corrosion depths as functions of time in three types of environment, which were obtained on the basis of three previous models:

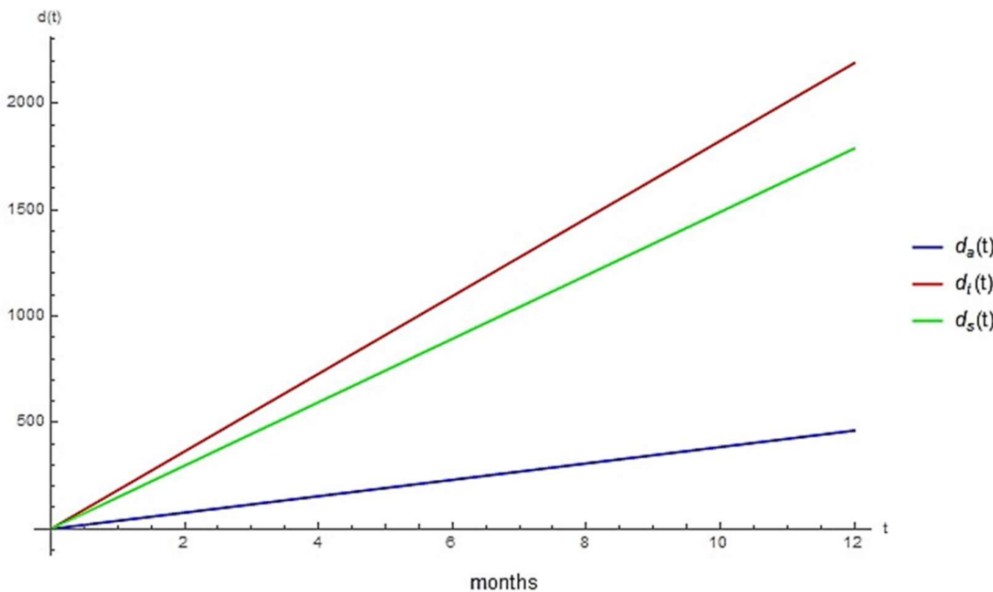

**Figure 7.** Corrosion depth expressed as a linear function of time.

The linear models presented confirmed that the corresponding corrosion rate equals $r_a(t) = d_a(t)' = 38.522 (\text{nm/month})$. This linear model was characterized by a *p*-value that equals 0.0000811097 and shows that the linear model describes the data from Table 1 properly. A similar conclusion was also reached for the rate of corrosion caused by the influences of flood tides and the sea. The rate of corrosion influenced by flood tides equals $r_t(t) = 182.43014$ (nm/month), while the rate of corrosion influenced by the sea, was $r_s(t) = 148.99187$ (nm/month).

Numerous factors with stochastic characteristics affect corrosion, and, therefore, the $c_1$ coefficient should be regarded as a continuous random variable rather than a constant, which is the central idea of the paper. The study proposes a probabilistic approach to the approximation of the Cumulative Distribution Function (CDF) of corrosion rate ($c_1$). The subsequent sections propose a methodology that relies on the statistical analysis of the CDF of $c_1$ coefficient, which is considered a random variable. With the assumption that the linear model $d(t) = c_1 t$ is acceptable and statistically correct, the empirical CDF of the $c_1 = \frac{d(t)}{t}$ coefficient can be determined, and, subsequently, two-parameter distributions can be fitted to these empirical values. The best fitted two-parameter distributions that describe the corrosion rate values are obtained in this way.

Each of the 27 most frequently used two-parameter distributions (Table 6) was analyzed for every type of environment (the air, tides, and sea).

**Table 6.** List of tested two-parameter continuous distributions.

|  | Distribution | Parameters |
|---|---|---|
| 1 | Cauchy | $\sigma, \mu$ |
| 2 | Chi-Squared (2P) | $\nu, \gamma$ |
| 3 | Erlang | $m, \beta$ |
| 4 | Exponential (2P) | $\lambda, \gamma$ |
| 5 | Fatigue Life | $\alpha, \beta$ |
| 6 | Frechet | $\alpha, \beta$ |
| 7 | Gamma | $\alpha, \beta$ |
| 8 | Gumbel Max | $\sigma, \mu$ |
| 9 | Gumbel Min | $\sigma, \mu$ |
| 10 | Hypersecant | $\sigma, \mu$ |
| 11 | Inverse Gaussian | $\lambda, \mu$ |
| 12 | Laplace | $\lambda, \mu$ |
| 13 | Levy (2P) | $\sigma, \mu$ |
| 14 | Log-Gamma | $\alpha, \beta$ |
| 15 | Log-Logistic | $\alpha, \beta$ |
| 16 | Logistic | $\sigma, \mu$ |
| 17 | Lognormal | $\sigma, \mu$ |
| 18 | Nakagami | $m, \Omega$ |
| 19 | Normal | $\sigma, \mu$ |
| 20 | Pareto | $\alpha, \beta$ |
| 21 | Pareto 2 | $\alpha, \beta$ |
| 22 | Pearson 5 | $\alpha, \beta$ |
| 23 | Rayleigh (2P) | $\sigma, \gamma$ |
| 24 | Reciprocal | $a, b$ |
| 25 | Rice | $\nu, \sigma$ |
| 26 | Uniform | $a, b$ |
| 27 | Weibull | $\alpha, \beta$ |

The statistical analysis whose results were verified by means of the Kolmogorov–Smirnov test was conducted in order to determine the three best fitted two-parameter distributions that describe the empirical data obtained on the basis of the measurements of corrosion rate coefficient $(c_1)$ adequately. The corrosion rate coefficient is considered to be a function of time and is affected by environmental factors. The main advantage of the Kolmogorov–Smirnov test is the fact that it is non-parametric and distribution-free. The test is also applicable in cases when it should be detected if a sample originated from a continuous distribution. Null and alternative hypotheses for the Kolmogorov–Smirnov test are defined in the following way:

$H_0$—the data follow the specified theoretical distribution,
Ha—the data do not follow the specified theoretical distribution.

The goodness of fit shows how well a selected probability function fits the data measured. The comparison between a calculated p-value and a critical value for predefined significance level was made in order to check the quality of the distribution fitted. Table 7 shows the values of significance level $\alpha$ and the corresponding critical values that were used in the paper.

**Table 7.** Values used for significance level and corresponding values for critical value.

| $\alpha$ | 0.2 | 0.1 | 0.05 | 0.02 | 0.01 |
|---|---|---|---|---|---|
| Critical Value | 0.16547 | 0.18913 | 0.21012 | 0.23494 | 0.25205 |

The hypothesis regarding the distributional form was rejected for the chosen $\alpha$ significance level if the test statistic exceeded the critical value from Table 7. The p-value was

calculated based on the test statistics and denoted the threshold value of a significance level. A null hypothesis would be accepted for all values of $\alpha$ that are lower than the p-value. The null hypothesis was rejected if the p-value was lower than the selected critical value, and it can be concluded that the theoretical distribution did not describe the empirical data for the selected significance level. Generally, if the calculated value of the test statistic was low, there was not a significant statistical difference between the theoretical and empirical values. More precisely, the calculated values of the test statistic should be lower than the critical value selected.

The analyzed two-parameter distributions depend on the set of different parameters ($\theta$) shown in Table 6. Theoretical distributions were defined on the basis of the Probability Density Functions (PDF). PDF was also used to determine formulas for the CDF of the observed theoretical distributions. In the following sections, CDF will be labeled as $F(x)$, while PDF will be labeled as $f(x)$.

The fitting of theoretical distributions to the set of data obtained through the measurements of corrosion rate in different types of environment (the air, flood tides, and the sea) aims to determine the value of the set of $\theta$ parameters of theoretical distributions. Each theoretical distribution should describe the distribution of empirical data adequately for each type of environment examined (the air, flood tides, and the sea). The corresponding set of $\theta$ theoretical parameters for each two-parameter distribution was determined by the maximum likelihood estimation method.

According to the Kolmogorov–Smirnov test, Weibull, Normal, and Nakagami distributions were the three best two-parameter distributions fitted to the data that represented the depth of corrosion influenced by the air. Figure 8 shows the graph of the PDF of best-fitted two-parameter distributions.

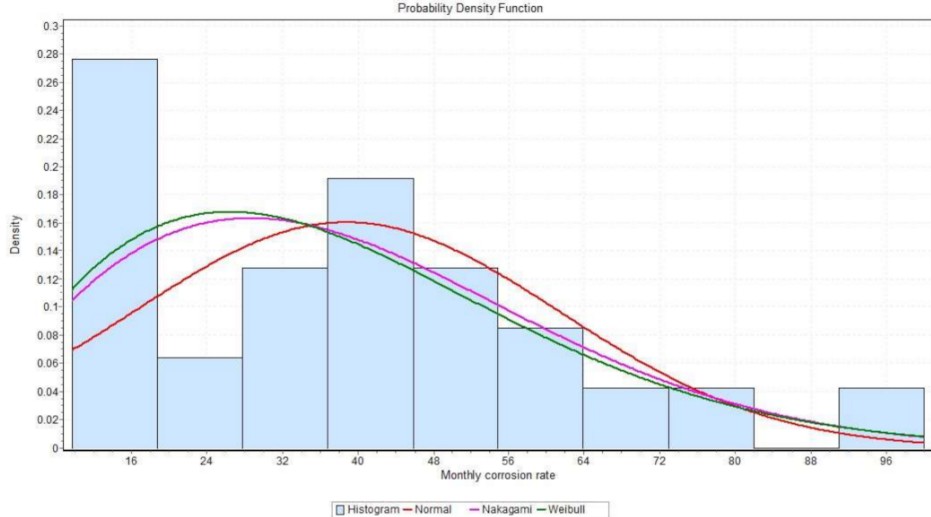

**Figure 8.** Probability density function (PDF) graphics for the three best fitted two-parameter distributions for air influence.

The expressions for fitted PDF (denoted as $f_a^{we}(x)$) and CDF (denoted as $F_a^{we}(x)$) of the Weibull distribution of $c_1$ in the case of air, influence are, respectively, given by

$$f_a^{we}(x) = 0.0024 \mathrm{e}^{-0.0014x^{1.7551}} x^{0.755} \text{ for } x \geq 0, \tag{5}$$

$$F_a^{we}(x) = 1 - \mathrm{e}^{-0.0014x^{1.755}}. \tag{6}$$

The corresponding mean and Standard Deviation of the coefficient $c_1$ are, respectively, equal to 38.046 (nm/month) and 22.378 (nm/month). It is worth noting that the previous mean value was very close to the corrosion rate $r(t) = 38.522$ (nm/month) obtained based on the linear fitted model for $d(t)$ given by Equation (1).

The expressions for fitted PDF (denoted as $f_a^{no}(x)$) and CDF (denoted as $F_a^{no}(x)$) of Normal distribution of $c_1$ in the case of air, influence are, respectively, given by

$$f_a^{no}(x) = 0.01e^{-0.0003}(x - 22.445)^2 \text{ for } -\infty < x < +\infty, \quad (7)$$

$$F_a^{no}(x) = 0.01 \int_{-\infty}^{x} e^{-0.0003}(t - 22.445)^2 dt. \quad (8)$$

The corresponding mean and Standard Deviation of the coefficient $c_1$ are equal to 38.799 and 22.445 (nm/month), respectively. The mean value was again very close to the corrosion rate $r(t) = 38.522$ (nm/month) obtained based on the linear fitted model for $d(t)$ given by Equation (1).

Nakagami distribution was ranked as the third best two-parameter distribution. Its PDF ($f_a^{na}(x)$) and CDF ($F_a^{na}(x)$) of coefficient $c_1$ in the case of air, influence are, respectively, given by

$$f_a^{na}(x) = 0.0025e^{-0.0004x^2}x^{0.699} \text{ for } x \geq 0, \quad (9)$$

$$F_a^{na}(x) = 0.00256 \int_{-\infty}^{x} e^{-0.0004t^2}t^{0.699} dt = \int_{0}^{0.0004\,x^2} t^{-0.15}e^{-t} dt. \quad (10)$$

The corresponding mean and Standard Deviation of the coefficient $c_1$ are equal to 38.839 and 22.135 (nm/month), respectively. It can be concluded that the calculated mean value was very close to the corrosion rate $r(t) = 38.522$ (nm/month) obtained based on the linear fitted model.

In terms of the influences of flood tides on corrosion, Fatigue life, Lognormal, and Log-logistic distributions are considered as the three best fitted two-parameter distributions, according to the Kolmogorov–Smirnov test. Figure 9 shows the graph of the PDF of the listed distributions.

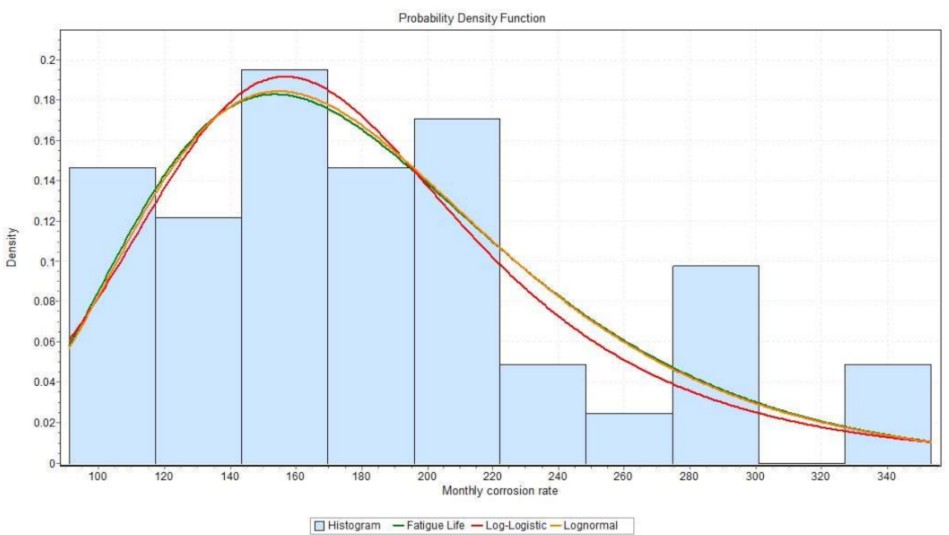

**Figure 9.** PDF graphics for the three best fitted two-parameter distributions for air influence

Fatigue life distribution has the following analytical forms for PDF and CDF, respectively:

$$f_t^{fl}(x) = \frac{e^{-\frac{0.02}{x} - 710.41x}(149.56 + 26070.56x)}{x^{\frac{3}{2}}} \text{ for } x > 0, \quad (11)$$

$$F_t^{fl}(x) = \frac{1}{2} + 4.31 \times 10^{-21}\left(-1.16 \times 10^{20}\frac{1}{\sqrt{\pi}}\int_{0}^{\frac{0.153}{\sqrt{x}} - 26.65\sqrt{x}} e^{-t^2} dt - 279.179\left(-1.20 \times 10^7 + 1.20 \times 10^7\frac{1}{\sqrt{\pi}}\int_{0}^{\frac{0.153}{\sqrt{x}} + 26.65\sqrt{x}} e^{-t^2} dt\right)\right). \quad (12)$$

Lognormal distribution je was ranked as the second best two-parameter distribution in the case of the influence of tides on corrosion processes. The PDF and CDF for the Lognormal distribution are given in the following terms, respectively:

$$f_t^{ln}(x) = \frac{0.077 e^{-0.019(-0.346+\ln x)^2}}{x} \text{ for } x > 0, \tag{13}$$

$$F_t^{ln}(x) = \frac{1}{2}\left(1 - \frac{1}{\sqrt{\pi}}\int_0^{0.14(0.346-\ln x)} e^{-t^2}dt\right). \tag{14}$$

The following analytic forms of PDF and CDF could be derived for the influences of flood tides in case of Log-logistic distribution:

$$f_t^{ll}(x) = \frac{9.77\times 10^{-11}x^{3.785}}{(1+2.04\times 10^{-11}x^{4.785})^2} \text{ for } x \geq 0, \tag{15}$$

$$F_t^{ll}(x) = \frac{1}{1+\frac{4.897\times 10^{10}}{x^{4.7851}}}. \tag{16}$$

Mean values for Fatigue, Lognormal, and Log-Logistic distribution are 185.0, 185.22, and 184.36 (nm/month), while.

Standard Deviation values are 65.568, 66.052, and 76.835 (nm/month), respectively. Notice that mean values for the listed three best two-parameter distributions are very close to the corrosion rate calculated based on the linear model given by Equation (2).

The fitting of two-parameter distributions to the empirical data obtained through the measurement of corrosive processes influenced by the sea, along with the data rating based on the Kolmogorov–Smirnov test, indicate that the three best fitted two-parameter distributions are GumbelMin, Weibull, and Inverse Gaussian. Figure 10 shows the graph of the PDF of these theoretical distributions.

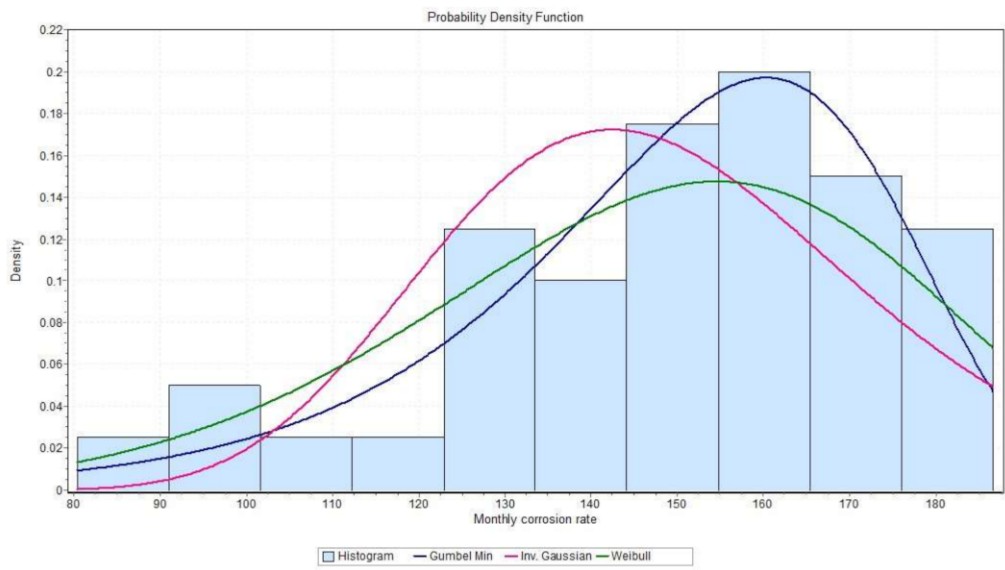

**Figure 10.** PDF graphics for the three best fitted two-parameter distributions for sea influence

PDF and CDF for GumbelMin, Weibull, and Inverse Gaussian distribution, denoted by $f_s^{gm}(x)$, $F_s^{gm}(x)$, $f_s^{we}(x)$, $F_s^{we}(x)$, and $f_s^{ig}(x)$, $F_s^{ig}(x)$ are respectively

$$f_s^{gm}(x) = 0.05e^{(0.05(x-160.32)-e^{0.05(x-160.32)})} \text{ for } -\infty < x < +\infty, \tag{17}$$

$$F_s^{gm}(x) = 1 - e^{-e^{0.05(x-160.32)}}, \tag{18}$$

$$f_s^{we}(x) = 5.43 \times 10^{-13} e^{-9.17 \times 10^{-14} x^{5.92}} x^{4.92} \text{ for } x \geq 0, \tag{19}$$

$$F_s^{we}(x) = 1 - e^{-9.17 \times 10^{-14} x^{5.92}}, \tag{20}$$

$$f_s^{ig}(x) = 4.87 e^{-\frac{0.0000029(-5084.6+x)^2}{x}} \sqrt{\frac{1}{x^3}} \text{ for } x \geq 0, \tag{21}$$

$$F_s^{ig}(x) = \frac{1}{2} \left( \begin{array}{c} 1 - \frac{1}{\sqrt{\pi}} \int_0^{-\frac{0.0017(x-5084.6)}{\sqrt{x}}} e^{-t^2} dt + 1.06 \times \\ \left( 1 - \frac{1}{\sqrt{\pi}} \int_0^{\frac{0.0017(5084.6+x)}{\sqrt{x}}} e^{-t^2} dt \right) \end{array} \right). \tag{22}$$

Mean values for GumbelMin, Weibull, and Inverse Gaussian distribution are 148.86, 147.83, and 148.85 (nm/month), while Standard

Deviation values are 25.469, 29.007, and 25.468 (nm/month), respectively. Notice that mean values for the listed three best two-parameter distributions are very close to the corrosion rate calculated based on the linear model given by Equation (3).

Figure 11 shows the graphic view of the CDF of the three best fitted theoretical two-parameter distributions for the three types of environment examined.

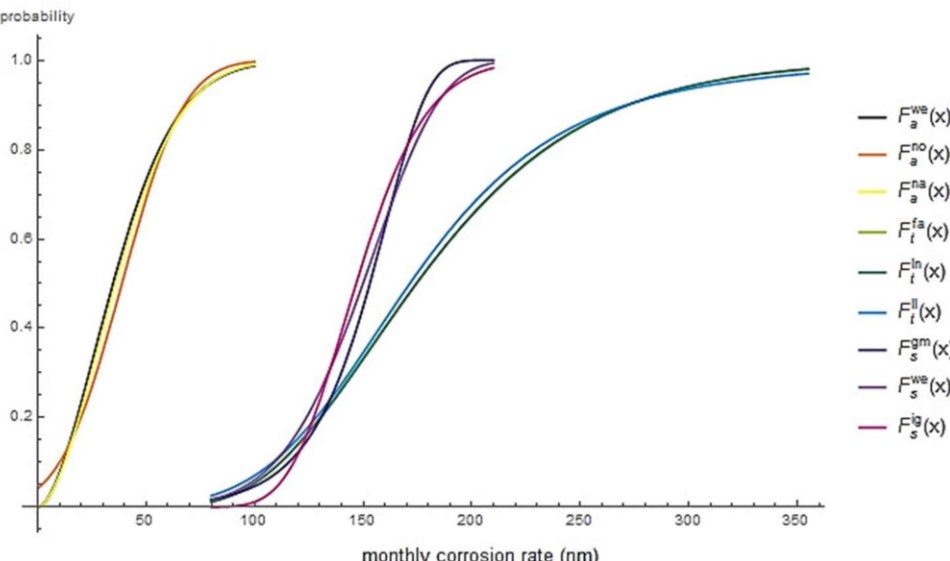

**Figure 11.** Comparative presentation of CDF graphs for the three best fitted two-parameter distributions for the influence of three different seawater environments.

Tables 8–10 show the three best-fitted distributions that were obtained through the ratings according to the Kolmogorov–Smirnov test for the influences of the air, tides, and the sea on corrosion. The values of test statistics and *p*-value were calculated for each theoretical distribution based on the Kolmogorov–Smirnov test. The Kolmogorov–Smirnov test statistic was calculated as the largest vertical difference between theoretical and empirical CDFs. The rating of two-parameter distributions was organized in a non-descending array of test statistic values and in a non-ascending array of *p*-values. The two-parameter distribution with the lowest value of test statistic is the best fitted theoretical distribution. Therefore, it can be concluded that a two-parameter distribution with the lowest value of test statistic best describes empirical data. On the contrary, the best fitted two-parameter distribution was characterized by the highest *p*-value.

**Table 8.** Kolmogorov–Smirnov test values for the three best fitted two-parameter distributions for the air influence.

| AIR: Kolmogorov–Smirnov Test Ranking | | | |
|---|---|---|---|
| Distribution | 1. Weibull | 2. Normal | 3. Nakagami |
| Statistic | 0.09944 | 0.10073 | 0.10282 |
| *p*-value | 0.70408 | 0.68916 | 0.66495 |

**Table 9.** Kolmogorov–Smirnov test values for the three best fitted two-parameter distributions for the tide influence.

| TIDE: Kolmogorov–Smirnov Test Ranking | | | |
|---|---|---|---|
| Distribution | 1. Fatigue Life | 2. Lognormal | 3. Log-Logistic |
| Statistic | 0.07773 | 0.07828 | 0.0783 |
| *p*-value | 0.94906 | 0.94621 | 0.94612 |

**Table 10.** Kolmogorov–Smirnov test values for the three best fitted two-parameter distributions for the sea influence.

| SEA: Kolmogorov–Smirnov Test Ranking | | | |
|---|---|---|---|
| Distribution | 1. Gumbel Min | 2. Weibull | 3. Inv. Gaussian |
| Statistic | 0.0821 | 0.10442 | 0.11332 |
| *p*-value | 0.92996 | 0.73665 | 0.64192 |

The comparison between the values of test statistics in Tables 8–10 and the predefined critical values for the selected significance level shown in Table 6 indicates that a null hypothesis was not rejected for the analyzed values of $\alpha$. Moreover, a null hypothesis was not rejected for either of the proposed best fitted theoretical distributions for the influences of the air, flood tides, and the sea.

*3.3. The Result of Different Sea Water Environment's Influence on CuAlNi*

In this manuscript, in addition to statistical analysis, a multivariate analysis, PCA, was applied to the EDX results, with the aim of obtaining information about the influence of the environment and exposure time on the corrosion behavior of the CuAlNi.

The registration and distinction of the different influences of various types of the environment on the degradation of metals can be performed through the application of PCA and in accordance with the EDX results, as confirmed in the previous study. The analysis of the CuAlNi detected similar trends in corrosive behavior in different types of environments. Figure 12 shows the corrosive behavior of the alloy based on the EDX results after six months of exposure to various types of the corrosive environment.

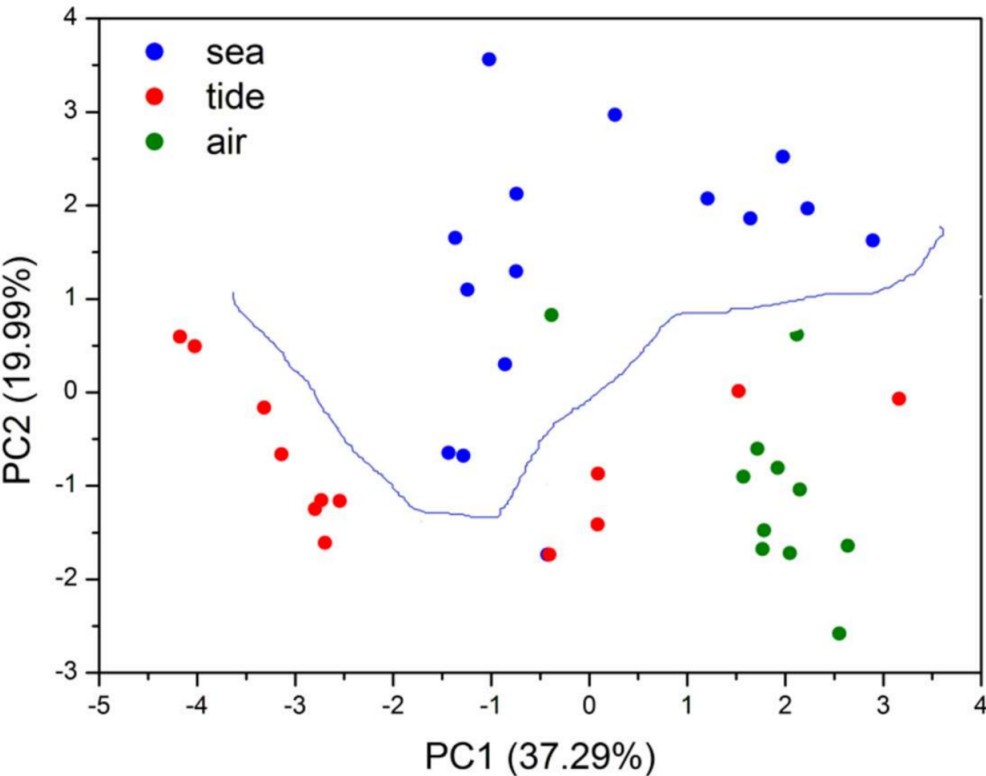

**Figure 12.** Score plot as a result of PC1 versus PC2 (after 6 months).

The obtained PCA results confirm that the applied multivariate method of analysis could detect large and small differences in the corrosion behavior of this alloy after 6 months as a function of different environments. As Figure 12 shows, two principal components (PC1–PC2) give a very good description of the corrosion behavior (with very small deviations) of the alloy, depending on the characteristics of the environment the alloy was exposed to. As Figure 12 illustrates, the PCA, with equal precision, registered the existence of a difference between the corrosion behavior of the alloy when exposed to flood tides (T) and the air (A), as well as the notably different influence of the sea (S) on CuAlNi dissolution.

Namely, the corrosive behavior of the CuAlNi samples in the sea was characterized by the positive values of PC2, while the influence of the air resulted in positive PC1 values and negative PC2 values. According to the PCA analysis, the samples exposed to the influences of flood tides were characterized by negative PC1 and negative PC2 values, which differentiates the samples affected by the flood tides from the samples affected by the air.

Furthermore, the study examined the possibility of the applied multivariate analysis to register the influence of the length of exposure of the alloy to the different types of the environment on the degree of damage.

Figure 13 shows the correlation between PC1 and PC2, which were obtained through the PCA analysis of the data from the EDX analysis of the CuAlNi samples after 12 months of exposure to the types of environment examined.

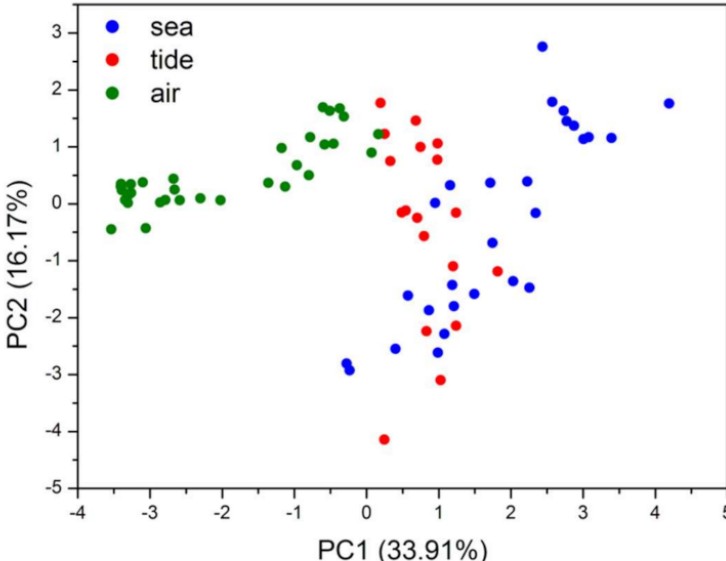

**Figure 13.** Score plot as a result of PC1 versus PC2 (after 12 months).

After 12 months, a slightly different image can be registered (Figure 13). As with the measurements performed after 6 months under these conditions, clear separation of samples exposed to the air can also still be observed, while between samples exposed to sea and tides, the noticeable separation that existed after 6 months was lost, and data largely overlapped. This distribution of the obtained results may indicate that the corrosion of CuAlNi in the sea and in the air continues the trend of behavior regardless of time, while in the case of flood tides, the mechanism of dissolution of the CuAlNi changed as a function of time.

The data obtained through the EDX analysis were examined by means of the PCA method for each type of environment on the basis of a matrix that contained the data for the environment examined in both time intervals (6 and 12 months). This examination was performed in order to provide more precise data on the specific influences of the particular types of the environment on the behavior of the CuAlNi over the time interval examined.

Figure 14 shows the correlation between PC1 and PC2, which was obtained through the analysis of the samples that were exposed to the influences of the sea (a), flood tides (b), and the air (c) over the time interval indicated.

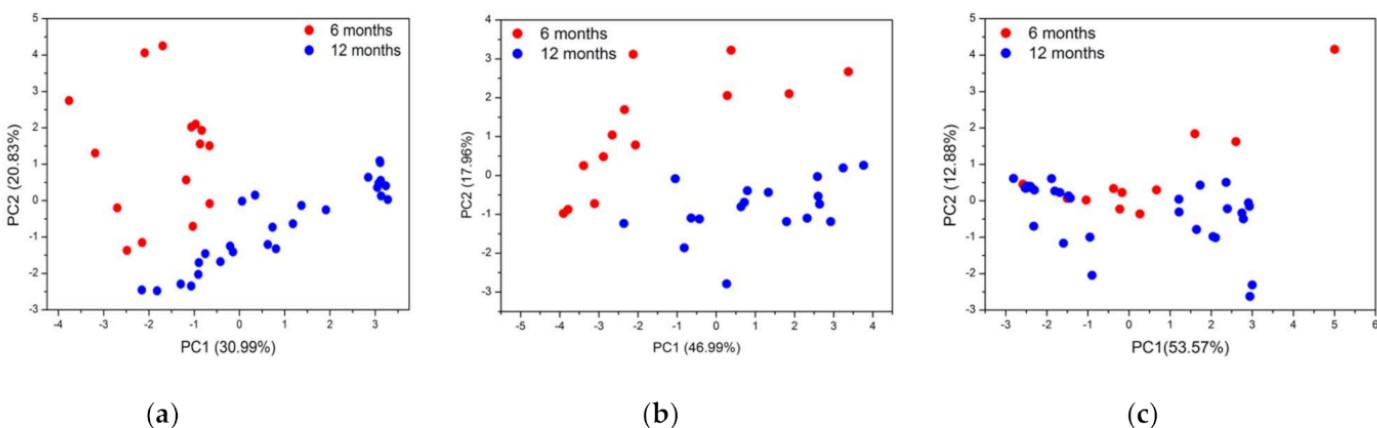

**Figure 14.** Score plot (PC1 vs. PC2) after both time intervals exposure to (**a**) sea, (**b**) flood tides, (**c**) air.

Figure 14 shows clearly the various effect of the examined environment on the corrosive behavior of the CuAlNi over the time interval examined. The sea (Figure 14a) and flood tides (Figure 14b) had the same influence on the corrosive behavior of the examined

alloy. Namely, in both environments, there was a clear separation of the tested alloy on the basis of corrosion behavior after 6 and 12 months. The sample that was exposed to the influence of the air exhibited different trends in corrosive behavior. There was not a significant difference between the data obtained after 6 and after 12 months, as shown in Figure 14c. These effects of the marine environment were related to considerably slower corrosion of the CuAlNi exposed to the influences of the air. Contrary to the influences of the sea and flood tides, the additional 6 months of exposure to the air did not cause significant corrosive damage of the CuAlNi.

## 4. Conclusions

The main findings are as follows:

Empirical data analysis of the marine environment's influence on the corrosion processes of CuAlNI SMA represented by a linear model showed that the mean value of monthly corrosion for air, tides, and sea, were formatted corrosion layers with depths of 38.522, 182.43014, and 148.99187 nm, respectively. The linear model had shown clearly that the tides had the greatest influence on the corrosion processes, followed by the sea, while the air had the lowest influence.

There were 27 continuous two-parameter distributions tested and based on the statistics obtained by the Kolmogorov–Smirnov test, it can be concluded that the best fitted two-parameter distribution for the air's influence is the Weibull distribution, for the tides' influence Fatigue life distribution, and Gumbel Min distribution for the sea's influence. The significance level of statistical analysis showed that all three proposed best two-parameter continuous distributions followed the empirical data describing the corrosion processes in all three environments (air, tides, sea) adequately.

The graphic analysis showed that the corrosion rates influenced by the air and sea had a similar curve shape. The similar character of these corrosion values can be explained, bearing in mind the calculated Standard Deviations ranging between 22.135–22.445 for air and 25.468–29.007 for the sea, although the influence of the sea on the corrosion processes is, in principle, greater than the influence of the air. On the other hand, the corrosion rate of the tides was the fastest because the thickest corrosion layer had formed (mean was 182.43 nm); the rapid increase of the corrosion rate was connected with the longer process time. In addition, there was a significantly greater dispersion of results calculated in the tides' environment (SD was between 65,568 and 76,835), which can be attributed to the greater tide dynamics, and, with that, major changes in the chemical/physical parameters.

The final conclusion is that PCA analysis registered the boundaries between the influence of the sea, tides, and atmosphere on the corrosion behavior of the CuAlNI SMA clearly. Furthermore, the PCA registered that the weather changes in the time of 6 and 12 months had different effects on the corrosion process of the CuAlNi SMA, depending on the environment to which they were exposed. It was found out that the additional 6 months' exposure of CuAlNI SMA to the air did not cause significant additional corrosive damage of CuAlNi SMA, which is contrary to the influences of the sea and flood tides.

**Author Contributions:** Conceptualization, Š.I., G.V. and R.R.; methodology, Š.I., G.V. and R.R.; validation, Š.I., G.V. and R.R.; formal analysis, P.M. and N.K.; investigation, Š.I., G.V., P.M. and R.R.; resources, Š.I. and R.R.; data curation, N.K. and G.V.; writing—original draft preparation, Š.I., G.V. and R.R.; writing—review and editing, Š.I. and R.R.; visualization, P.M.; supervision, R.R.; project administration, Š.I. and R.R.; funding acquisition, Š.I. and R.R. All authors have read and agreed to the published version of the manuscript.

**Funding:** This research was funded by the Bilateral projects Slovenia—Montenegro (BI-ME/18-20-024) and Serbia—Montenegro and Eureka program PROCHA-SMA E!13080 funded by Ministry of Science of the Republic of Montenegro.

**Institutional Review Board Statement:** Not applicable.

**Informed Consent Statement:** Not applicable.

**Data Availability Statement:** Not applicable.

**Acknowledgments:** This paper is a result of the initial phase of the research of different aspects of the sea and atmosphere to the production and application of smart materials of Shape Memory Alloy in the Nautical industry. Project PROCHA-SMA is a part of the EUREKA Project, which is jointly realised by the Faculty of Stomatology in Belgrade, Zlatarna Celje and the Faculty of Maritime Studies Kotor, University of Montenegro.

**Conflicts of Interest:** The authors declare no conflict of interest.

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
