# Peer review of "A Probabilistic Method for Estimating the Influence of Corrosion on the CuAlNi Shape Memory Alloy in Different Marine Environments"

_crystals, doi:10.3390/cryst11030274_

Round 1

Reviewer 1 Report

Comments for authors are in the attachment.

Reviewer 2 Report

The work layout requires correction:

Introduction

Lines 165- 174 should be in Materials and Methods chapter

Materials and Methods

Lines 209- 217 should be in Introductions chapter

Lines 218- 221 is a repetition of information from Introduction

Line 204 – 206 should be in Materials section

2.3. 2. EDX results should be in Results chapter

There is a very little information about ICP, XRF, EDX methods. Please add a discussion of the lower limits of detection and the errors involved. Add a few sentences explaining why you think these results from ICP, XRF, EDX are accurate, repeatable, and precise and can be used.  Where is the literature review of usage of your type of analyses?

References should be edited e.g. [11]

Round 2

Reviewer 1 Report

Accept.